# Precision Medicine of Hepatobiliary and Pancreatic Cancers: Focusing on Clinical Trial Outcomes

**DOI:** 10.3390/cancers14153674

**Published:** 2022-07-28

**Authors:** Takehiko Tsumura, Keitaro Doi, Hiroyuki Marusawa

**Affiliations:** 1Department of Medical Oncology, Osaka Red Cross Hospital, Osaka 543-8555, Japan; kdoi0160@osaka-med.jrc.or.jp; 2Department of Gastroenterology, Osaka Red Cross Hospital, Osaka 543-8555, Japan; maru@kuhp.kyoto-u.ac.jp

**Keywords:** precision medicine, next-generation sequencing, hepatobiliary cancer, pancreatic cancer, liquid biopsy

## Abstract

**Simple Summary:**

Recently, tumor-agnostic precision medicine employing comprehensive genome profiling (CGP) and using next-generation sequencing (NGS) has been progressing. This review focuses on the recent progress in precision medicine for advanced unresectable hepatobiliary and pancreatic cancers. We highlight therapies that target several regulators of cancer cell growth and tumor-agnostic therapies that effectively treat biliary and pancreatic cancers. We also discuss our current understanding of precision medicine methods developed using NGS of circulating tumor DNA (ctDNA) as a liquid biopsy technique.

**Abstract:**

Tumor-agnostic precision medicine employing comprehensive genome profiling (CGP) and using next-generation sequencing (NGS) has been progressing recently. This review focuses on precision medicine for advanced unresectable hepatobiliary and pancreatic cancers. In this paper, for biliary tract cancer (BTC), therapies that target several regulators of cancer cell growth, including isocitrate dehydrogenase 1 (IDH1), fibroblast growth factor receptor 2 (FGFR2) fusion, proto-oncogene B-Raf (BRAF), and human epidermal growth factor receptor 2 (HER2) alterations, are reviewed. For pancreatic ductal adenocarcinoma (PDAC), therapies for Kirsten rat sarcoma virus (KRAS) gene mutation G12C, neuregulin (NRG)1, and breast cancer type 1 and 2 susceptibility (BRCA1/2), gene alterations are summarized. On the other hand, precision medicine targets were not established for hepatocellular carcinoma (HCC), although telomerase reverse transcriptase (TERT), tumor protein P53 (TP53), and Wnt/β catenin signaling alterations have been recognized as HCC driver oncogenes. Tumor-agnostic therapies for microsatellite instability-high (MSI-H) and neurotropic tyrosine receptor kinase (NTRK) fusion cancers effectively treat biliary and pancreatic cancers. Precision medicine methods developed using NGS of circulating tumor DNA (ctDNA) and utilizing a liquid biopsy technique are discussed.

## 1. Introduction

Cancer is a genetic disease caused by a stepwise accumulation of genetic alterations. Revolutionary high-throughput sequencing capacity of next-generation sequencing (NGS) technologies supported a breakthrough in cancer genome research and accelerated the innovation in anti-cancer drug development. Several genetic alterations have been discovered in human cancerous tissues using NGS-based whole-exome and whole-genome sequencing. Precision medicine involves comprehensive genome profiling (CGP) using NGS, which is clinically performed worldwide to find a drug suitable for each patient. For example, in bile duct cancers, genetic aberrations in tumor protein P53 (TP53); Kirsten rat sarcoma virus (KRAS); phosphatidylinositol-4,5-bisphosphate 3-kinase, catalytic subunit alpha (PI3KCA); BAP1; cyclin-dependent kinase inhibitor (CDKN)2A/B; AT-rich interaction domain 1A (ARID1A); receptor tyrosine-protein kinase erbB-2 (ERBB2); proto-oncogene B-Raf (BRAF); fibroblast growth factor receptor (FGFR)1–3; isocitrate dehydrogenase (IDH)1/2; MET proto-oncogene, receptor tyrosine kinase (MET); and FGFR2 are frequently detectable and used as major markers for selective therapies [1,2]. In pancreatic cancers, frequent mutations of several cancer-related genes, such as KRAS, TP53, CDKN2A, SMAD family member 4 (SMAD4), CDKN2B, ARID1A, GATA6, and MYC, have been determined using NGS [3]. Additionally, in hepatocellular carcinoma (HCC), genetic alterations have been found in telomerase reverse transcriptase (TERT), p53/RB (TP53 and CDKN2A), or Wnt/βcatenin (CTNNB1 and AXIN1) signaling pathways [4,5]. The aim of precision medicine is to identify and target those genome-wide alterations in various cancers, and the genetic changes identified by NGS technologies support personalized drug design for patients with specific genetic alterations. Of note, a precision medicine approach is advanced in lung adenocarcinoma. For instance, osimertinib, a third-generation tyrosine kinase inhibitor, prolonged progression-free survival (PFS) to 18.9 months in patients with lung adenocarcinoma with the epidermal growth factor receptor (EGFR) common mutation [6]. Another example is malignant melanoma with BRAF mutation. BRAF and MEK inhibitors, dabrafenib, and trametinib combination therapy prolonged PFS and overall survival (OS) to 12.3 and 22.3 months, respectively [7]. Precision medicine-targeted genetic alterations and treatment results for hepatobiliary and pancreatic cancers, which will be discussed in this study, are summarized in Table 1. This review focuses on the recent progress in precision medicine for advanced unresectable hepatobiliary and pancreatic cancers.

## 2. Precision Medicine Targets in Biliary Tract Cancer (BTC)

BTC is a well-targeted cancer type addressed by precision medicine. The European Society for Medical Oncology precision medicine working group guidelines recommended that CGP should be investigated using NGS in this cancer [22]. Genomic findings differ for the three BTC types, including intrahepatic cholangiocarcinoma (IHCCA), extrahepatic cholangiocarcinoma (EHCCA), and gallbladder cancer (GBCA). Gene mutations of FGFR1–3, TP53, IDH1/2, ARID1/2, CDKN2A/B, and KRAS were found in IHCCA [1,2]. Changes in KRAS, TP53, and SMAD4 were detected in EHCCA [1,2]. Important alterations in TP53, ERBB2/3, CDKN2A/B, ARID1A, and KRAS have also been reported in GBCA [1,2]. These established precision medicine targets and outcomes of clinical trials for these cancer types are discussed below.

### 2.1. IDH1

IDH is an enzyme that regulates energy metabolism in the citric acid cycle. Wild-type IDH1 contributes to the reaction, producing α-ketoglutaric acid from isocitric acid. Mutant IDH1 produces D-2-hydroxyglutarate (D-2-HG), an oncometabolite. D-2-HG competitively inhibits IDH1 enzyme activity that depends on α-ketoglutaric acid, causing carcinogenesis [23]. About 16% of patients with IHCC have IDH1 mutations [1]. Ivosidenib is a novel small molecule inhibitor of mutant IDH1 protein. ClarIDHy is an international phase 3 study that proved the efficacy of ivosidenib against BTC with IDH1 mutation. PFS, which was the primary study endpoint, was 2.7 months (95% CI: 1.6–4.2 months) and 1.4 months (95% CI: 1.4–1.6 months) for ivosidenib and placebo groups, respectively [8]. The study reported a hazard ratio (HR) of 0.37 (95% CI: 0.25–0.54; one-sided *p* < 0.0001). Thus, ivosidenib significantly prolonged PFS [8]. Mean survival time (MST), which was the secondary endpoint parameter for ivosidenib, was 10.3 months (95% CI: 7.8–12.4 months), while that of the placebo groups was 5.1 months (95% CI: 3.8–7.6 months; adjusted for crossover), with an HR of 0.49 (95% CI: 0.34–0.70; one-sided *p* < 0.001) [9]. Therefore, ivosidenib also significantly extended the MST of BTC patients with IDH1 mutation. Ascites was the most common grade 3 or worse adverse event in both treatment groups found in 7% (4/59) of patients receiving placebo and 7% (9/121) receiving ivosidenib. Serious adverse events occurred in 36% (36/121) of patients receiving ivosidenib and 22% (13/59) of the placebo groups.

### 2.2. Fibroblast Growth Factor Receptor (FGFR)2 Fusion

FGFs are growth factors comprising 23 family members, which can bind to FGFR1–4 as ligands. FGFR2 fusion genes are known as driver oncogenes [24,25]. The prevalence of FGFR2 mutations in IHCCA was reported as 5–15%. At least 19 fusion partners were identified and contained functional domains related to the formation of a dimer or multimer. Generated from the fusion gene, FGFR2 forms a dimer which is activated without ligand stimulation, causing oncogenesis [24,25].

Pemigatinib selectively inhibits FGFR1–3, and its efficacy against BTC with a treatment history was indicated in a single-arm open-label phase 2 FIGHT-202 study. In the study, cohort A was composed of 107 patients with FGFR2 fusion/rearrangement, cohort B included 20 patients with FGF/FGFR2 mutation, and cohort C comprised 18 patients without FGF/FGFR2 alteration. Cohort A contained 56 fusion-partner genes, and BICC1 was the most frequently found. The objective response rate (ORR) of cohort A, the primary endpoint, was 35.5% (95% CI: 26.5–45.4%). The ORRs of cohorts B and C, which were secondary endpoints, were 0%. The disease control rate (DCR) of cohorts A, B, and C were 82% (95% CI: 74–89%), 40% (95% CI: 19–64%), and 22% (95% CI: 6–48%), respectively. The median PFSs in A/B/C cohorts were 6.9 months (95% CI: 6.2–9.6 months), 2.1 months (95% CI: 1.2–4.9 months), and 1.7 months (95% CI: 1.3–1.8 months), respectively. The median survival times were 21.1 months (95% CI: 14.8 to not estimated months), 6.7 months (95% CI: 2.1–10.6 months), and 4.0 months (95% CI: 2.3–6.5 months), respectively, for those cohorts. Grade 3 or worse treatment-related adverse events (trAEs) occurred in 71 patients (49%), including hypophosphatemia in 10 patients (7%) and stomatitis in 8 patients (5%) [10].

Infigratinib is another selective inhibitor of FGFR1–3, and the drug efficacy against BTC was tested in a multicenter open-label phase 2 study. Advanced BTCs with a treatment history with gemcitabine-containing regimen were registered. The number of registered participants included 48 patients (78.7%) with FGFR2 fusions, 8 patients (13.1%) with FGFR2 mutants, and 3 patients (4.9%) with FGFR amplifications. The observed ORRs, which was the primary endpoint, was 14.8% (95% CI: 7.0–26.2%) (18.8% for patients with FGFR2 fusion only). DCR was 75.4% (95% CI: 62.7–85.5%) (83.3% FGFR2 fusion only), and PFS was 5.8 months (95% CI: 4.3–7.6 months). As a notable grade 3 or 4 treatment-related adverse event, hyperphosphatemia occurred in 16.4% of the patients. [11].

These findings indicated that targeted therapy against FGFR2 alterations could be a promising precision medicine for BTC patients. Consistently, TAS-120 (futibatinib) is also being developed as a therapy for patients with FGFR2 fusion.

### 2.3. EGFR (HER)2 Amplification

HER2 (encoded by ERBB2 gene) is a membrane-penetrating receptor tyrosine-protein kinase. After the attachment of a specific ligand, the receptor forms a heterodimer and triggers growth-stimulating intracellular signaling. Overexpression or mutation of HER2 continuously activates the mitogen-activated protein kinase (MAPK) and PI3K/AKT pathways independently without a ligand. Therefore, HER2 overexpression/mutations are often associated with tumor proliferation or inhibition of apoptosis [26,27,28,29,30,31]. ERBB2 amplification was rarely found in IHCCA(3%) but was more common in EHCCA (11%) and GBCA (16%) [1].

The MyPathway basket trial verified the tumor-agnostic effect of trastuzumab and pertuzumab combination therapy in HER2 amplified/overexpressing cohort [12]. The ORR of all cancer types was 26% (95% CI: 19–35%) and DCR was 40% (46/114). The ORR of patients with BTC included in this study was 29% (2/7; 95% CI: 4–71%) and DCR was 86% (6/7). The efficacy of trastuzumab + pertuzumabut revealed in this study [12] were higher than ado-trastuzumab emtansine (T-DM1) [13], supporting the better anti-cancer quality of this approach.

The National Cancer Institute Molecular Analysis for Therapy Choice (NCI-MATCH) trial verified the effectiveness of T-DM1, an anti-body drug conjugate. HER2 overexpressed patients with more than seven copy numbers (confirmed by NGS) showed a PFS of 3.1 months (90% CI: 2.1–4.4 months). Although the ORR, which was the primary endpoint, was 5.6% (95% CI: 1.0–16.5%), it was not a satisfactory value. All three patients of BTC included in this study were diagnosed with stable disease (SD). Fatigue, anemia, fever, and thrombocytopenia were common toxicities, although no new safety problems were reported [13].

### 2.4. BRAF V600E

BRAF (A, B, and C subtypes) protein is a serine/threonine kinase associated with the MAPK pathway [32,33,34]. The binding of BRAF ligands forms a dimer, which transmits signals downstream of the MAPK network. Mutated BRAF could play a role as oncogenes. BRAF mutants activate downstream growth-promoting signals continuously, causing oncogenic transformation. The most common oncogenic BRAF mutation is BRAF V600E [35]. BRAF mutation frequency was reported at 5% in BTC [1]. BRAF V600E is one of the tumor-agnostic targets of precision medicine. The ROAR basket trial indicated that the combination therapy with dabrafenib (BRAF inhibitor) plus trametinib (MEK inhibitor) was effective against the BRAF V600E-mutated BTC [14]. Consequently, an independent reviewer-assessed overall response was 47% (95% CI: 31–62%). Increased γ-glutamyltransferase was the most common grade 3 or worse adverse event observed in five (12%) patients. Nine patients (21%) had treatment-related major adverse events, with pyrexia in eight (19%) being the most frequent [14].

## 3. Precision Medicine Targets in Pancreatic Ductal Adenocarcinoma (PDAC)

PDAC has a comparatively uniform genetic background comprising four mutated genes: KRAS, CDKN2A [p16/p14], TP53, and SMAD4 [3]. Although genetic alterations were frequently observed in ARID1A, KMT2C, RNF43, FAT3, and KMT2D [3], KRAS, BRCA2, and NRG1 are considered more valuable in cancer diagnostic using precision medicine for PDAC [3]. The results of clinical therapies, which targeted these three oncogenes in patients with PDAC, are described below.

### 3.1. BRCA1/2

PDAC represents a phenotype of hereditary breast and/or ovarian cancer syndrome, the well-known hereditary tumor syndrome caused by the BRCA1/2 mutation. The main phenotypes are breast and ovarian cancers, as its name suggests. Instances of prostate and other cancers have also been reported in carriers of the mutation [36,37,38]. BRCA mutation causes decompensation of double-strand DNA break repair—homologous recombination repair deficiency. Poly (ADP-ribose) polymerase (PARP) inhibitor, which inhibits single-strand DNA break compensation, can activate apoptosis in cancer cells with BRCA1/2 mutation and increase anti-cancer platinum sensitivity. However, only 6% of metastatic PDACs had BRCA1/2 mutations [15]. The POLO study showed the efficacy of olaparib against PDAC in patients with mutations that prevented progression for more than 16 weeks. The ORRs were 23% (18/78) and 12% (6/52) for olaparib and the placebo group, respectively. Although the OS, which was the secondary endpoint, was 18.9 and 18.1 months, respectively, and the hazard ratio (HR) was 0.91 (95% CI: 0.56–1.46, *p* = 0.68), there was no significant difference. The PFS, which was the primary endpoint, was 7.4 and 3.8 months, respectively, and the HR was 0.53 (95% CI: 0.35–0.82, *p* = 0.68) and was significantly different. Grade 3 or worse trAEs, including anemia (11%), fatigue or asthenia (5%), and decreased appetite (3%), occurred in 36 patients (40%) [16].

### 3.2. KRAS G12C

KRAS mutation is a significant oncogenic mutation and is found in 95% of PDACs. RAS is activated by connecting with GTP, receiving the signal from receptor tyrosine kinase. Activated RAS transmits the proliferation-promoting signal to the downstream MAPK pathway effectors. RAS, which has GTPase function, converts GTP into GDP via hydrolysis and proceeds to its inactivated state. However, mutated RAS has weak GTPase activity and induces a prolongation of activated signaling downstream, contributing to a malignant trait. Several efforts have been made to develop drugs that target mutated RAS. The effectiveness and safety of sotorasib, which affect mutated KRAS G12C, have been assessed in a phase 1/2 study named CodeBreaK 100 trial [18]. A total of 129 patients were included in both dose-escalation and enlarged cohorts; 59 patients with non-small cell lung cancer (NSCLC), 42 patients with colon cancer, and 28 patients with other cancers were registered, and all of them were included in the analysis. Generally, only 1.7% of PDACs had KRAS G12C mutation [17], and only 11 patients were registered. Other cancer groups, including PDAC cases, showed 14.3% (95% CI: 4.03–32.67%) of ORR and 75.0% (95% CI: 55.13–89.31%) of DCR. Although the value of ORR was low compared to that of NSCLC at 32.3% (95% CI: 20.62–45.64%), 9 of 11 patients with PDAC showed responses no less than SD, including one partial response; thus, the DCR was 82% (9/11) [18]. Grade 3 or 4 trAEs occurred in 15 patients (11.6%), including an increase in the alanine aminotransferase (ALT) level in 5.4%, diarrhea in 3.9%, and anemia in 3.1%. Other RAS-targeting drugs, including MRTX849 [39,40] and ARS-3248/JNJ-74699157 for KRASG12C [41], siG12D LODER for KRASG12D [42], and compound 3144 for Pan-KRAS [43], are under development.

### 3.3. Neuregulin (NRG)1

NRG1 is now gaining attention as a new target for precision medicine among KRAS-negative PDACs [44,45]. NRG1 protein is a cell adhesion molecule that belongs to the NRG family and affects the epidermal growth factor receptor (EGFR). The epidermal growth factor (EGF)-like domain of NRG1 regulates itself or nearby cells using HER2s or HER3/HER4 dimers. About 0.5% of PDAC have NRG fusion, and the specific fusion partners in PDAC are CDH1, ATP181, and VTCN1 [19]. The efficacy of zenocutuzumab, an NGR inhibitor, is proved in the phase 1/2 global open-label clinical trial (eNRGy) and the early access program against NSCLC and PDAC showing NRG1 fusion. The ORR of PDAC was 42% (5/12, 90% CI: 18–69%), with a DCR of 92% (11/12), which was a promising result [20]. TrAEs occurred in 59% of 157 patients, including diarrhea (20%), asthenia/fatigue (13%), nausea (10%), and infusion-related reactions (7%), and most trAEs were grade 1–2.

## 4. Possible Targets for Precision Medicine of Hepatocellular Carcinoma (HCC)

The accumulation of gene aberrations induced by several risk factors, such as alcohol, hepatitis virus infection, or steatohepatitis, causes hepatocarcinogenesis. Genetic alterations in TERT, TP53, or Wnt/βcatenin signaling pathway could play a role as a cancer driver for HCC. Genetic analysis of 755 HCC patients using NGS revealed genetic aberrations in 44% (TP53), 35% (CTNNB1), 31% (ARID1A), and 12% (MYC) of the assessed genes. However, these alterations have not been targeted by therapies and precision medicine has not been well-established in patients with HCC. Although some tumor-agnostic gene alterations, including changes in NTRK and MET, lead to precision medicine, the incidences of these genes are low (2–4% for NTRK and 2% for MET) [4,5].

### 4.1. Telomerase Reverse Transcriptase: TERT

TERT is one of the most common driver oncogenes of HCC with respect to cancer cell immortality. Every time cells divide, the recurrent arrangements at the edge of the chromosomes become shorter, and when the telomere length is shortened to a decided extent, the cells die. Cancer cells can be immortalized because of the high telomerase activity, which maintains the telomere length. TERT codes a telomerase subunit and affects the regulation of telomerase activity, causing the oncogenic progression of various cancers. HCCs have a 50% point mutation in the promotor area that regulates the TERT gene expression. Although drugs targeted for TERT treatment are being developed, their efficacy has not yet been established [4,5].

### 4.2. TP53

TP53 is one of the most effective tumor suppressing genes [46]. TP53 mutations are found in various human cancers [47,48]. Point mutations of the gene were found in ~40% of HCC. The protein product of TP53, p53, functions as a transcription factor, suppresses transcription genes concerned with cell proliferation, and is called the “Guardian Deity of Genome.” The inactivation of p53 leads to genome instability and accumulates genome abnormalities that lead to oncogenesis. However, precision medicine based on the TP53 mutation has not yet been well established [4,5].

### 4.3. Wnt/βcatenin Signaling Pathway

The Wnt/βcatenin signaling pathway is important for organogenesis, maintenance of stem cells, adjustment of cell proliferation, movement, and polarity [49]. Alterations of this pathway were found in 40–50% of HCCs [21]. Most of these changes are point mutations of CTNNB1 exon3 that are responsible for CTNNB1 transfer into the nucleus. The mutations can activate MYC and CTNNDI, causing the proliferation of HCC cells [21]. This pathway alteration is also strongly related to the immunological properties of the tumor and might lead to a response to immunological therapy [50].

## 5. Tumor-Agnostic Precision Medicine

Recently, the efficacy of the same drug against common genetic alterations in all types of cancer has been established. Here, we discuss microsatellite instability-high (MSI-H), tumor mutation burden-high (TMB-H), and NTRK fusions. Although there are several reports showing the possible tumor-agnostic precision medicines for hepatobiliary and pancreatic cancers, the evidence is not enough because of an insufficiency in the number of examined cases, and thus further examinations are awaited. The tumor-agnostic genetic alterations and treatment results for hepatobiliary and pancreatic cancers are summarized in Table 2. In addition to these treatments, it is noteworthy that the United States Food and Drug Administration (FDA) recently accelerated approval of tumor-agonistic therapy for solid tumors with BRAF mutation [51].

### 5.1. Microsatellite Instability-High in MSI-H Abnormality Phenotype

Mismatch repair deficiency (dMMR), a DNA repair deficiency caused by MLH1, MSH2, MSH6, or PMS2 mutation, and accumulation of gene mutations, leads to oncogenesis. dMMR tends to make errors in duplicating sequences, repeating a microsatellite sequence from one to several times. When dMMRs are widely found in the genome, the abnormality status is called MSI-High (MSI-H). There are two methods to investigate MSI status: polymerase chain reaction and NGS-based methods [59]. The incidence of MSI-H in the assessed 11 cancer types was 5%, and that in hepatobiliary and pancreatic cancers was ~2% [52]. The effectiveness of immune checkpoint inhibitors (ICIs) against MSI-H tumors has already been established. This effectiveness is due to the abundant expression of a mutation-associated neoantigen, which facilitates the recognition of cancer cells by the immune system, a process that is defined as cancer surveillance [60,61]. Conversely, Lynch syndrome should be excluded from the MSI-H patient cohort. Lynch syndrome is one of the best-known hereditary cancers; its phenotypes are colonic, endometrial, and urinary tract cancers [62]. The tumor-agnostic effectiveness of pembrolizumab against MSI-H tumors is proven by the KEYNOTE-158 trial. In this study, the ORR for 27 tumor types, including BTC and PDAC, were reported as 34.3 (80/233, 95% CI: 28.3–40.8%), and those of BTC and PDAC were reported as 40.9 (9/22, 95% CI: 20.7–63.6%) and 18.2% (4/22, 95% CI: 5.2–40.3%), respectively. TrAEs of pembrolizumab occurred in 151 patients (64.8%), and grade 3 or worse trAEs were seen in 34 patients (14.6%). Immune-related adverse events (irAE) are notable trAEs of ICIs. In this study, irAEs and infusion reactions occurred in 54 patients (23.2%). Hypothyroidism, hyper-thyroidism, colitis, and pneumonitis were the most prevalent irAEs. Pneumonitis (*n* = 3; 1.3%), severe skin reactions (*n* = 3; 1.3%), and hepatitis (*n* = 2; 0.9%) were classed as grade 3 irAEs [55].

### 5.2. Tumor Mutation Burden-High (TMB-H) Phenotypes

Accumulated mutations on somatic genes, tumor mutation burden (TMB), is defined as the number of single nucleotide variations per megabase [63]. Tumors with more than 10 mutations per megabase were defined as TMB-High (TMB-H) tumors [56]. TMB differs by cancer type [64], and the frequency of TMB-H cancers was reported at ~13% in 2589 cases of various cancers, with Bellini’s duct carcinoma at 4.0% (28/706) [53]. There is a correlation between TMB and ORR to ICI. PDAC is a cancer that shows the least TMB and ORR to ICI. Conversely, HCC shows moderate TMB and ORR [65]. An abundant expression of mutation-associated neoantigens contributes to the efficacy of ICI in TMB-H cancers, similar to MSI-H cancers. The tumor-agnostic efficacy of ICI on TMB-H cancers is shown in the KEYNOTE-158 trial. Although ORR, the primary endpoint, was 29.4% (30/102, 95% CI: 20.8–39.3%) for 10 tumor types, hepatobiliary and pancreatic cancers were not contained in the TMB-H cases in this study. Grade 3 or worse adverse events occurred in 16 (15%) patients. However, the efficacy of ICI on TMB-H cancers has not yet been established [66,67], and further evidence is needed.

### 5.3. Neurotropic Tyrosine Receptor Kinase: NTRK

The TRK receptor families TRK A/B/C are coded by NTRK genes and transmit the signaling into the cells concerned with differentiations and activities of peripheral and central neural cells. When NTRK1/2/3 are fused with other genes, TRK 1/2/3 protein activates itself continuously and keeps sending the signals, causing oncogenesis of the cell [68]. The incidences of NTRK fusion in various tumor types are 0.26% and for BTC and PDAC is 0.34% and 0.25%, respectively [54]. Integrated analyses of three phase 1/2 studies—NCT02122913, NCT02637687, and NCT02576431—although it contained few BTCs and PDACs, showed the efficacy of larotrectinib. The ORR, the primary endpoint, was 79% (121/153, 90% CI: 72–85%) for all tumor types. One of the two patients both in the BTC and PDAC subgroups showed response. Increased alanine aminotransferase in eight (3%) patients, anemia in six (2%), and decreased neutrophil count in five (2%) were the most common grade 3 or 4 trAEs [57]. The efficacy of entrectinib has also been proven in the integrated analysis of three studies: ALKA-372-001; STARTRK-1, which are phase 1 studies; and STARTRK-2, which is an international phase 2 basket trial. According to the analysis, entrectinib was administered to 54 patients with tumors with NTRK fusions. The ORR and median duration of response (mDOR) were 57% (31/54 90% CI: 43.2–70.8%) and 10 months (90% CI: 7.1 to not estimated), respectively. In the study, one patient with BTC and two of three with PDAC responded. Increased weight in seven (10%) and anemia in eight (12%) were the most prevalent grade 3 or 4 trAEs [58].

## 6. CGP by Liquid Biopsy

CGP by circulating tumor DNA (ctDNA) analysis is an established and clinically applied liquid biopsy technique [69,70]. Liquid biopsy, as a diagnostic procedure, has two breakthrough advantages as follows: (1) specimen collection procedure is relatively easy and provides real-time genetic information, and (2) turnaround time is short. Conversely, there are several negative points to remember. Firstly, when the amount of ctDNA is small, existing alterations might not be detected [71]. Secondly, the procedure is complicated by aging, which increases the incidence of false positives by clonal hematopoiesis of indeterminate potential (CHIP) [72,73]. The most commonly involved genes include DNMT3A, TET2, and ASXL1; however, other frequently mutated genes include TP53, JAK2, SF3B1, GNB1, PPM1D, GNAS, and BCORL1 [74,75,76]. Given the limited evidence, caution is needed when interpreting ctDNA variants in these genes. Finally, detecting copy number alterations or DNA fusions is occasionally difficult [69].

In biliary and pancreatic cancers with major genetic alterations, over a 90% concordance between histology and liquid biopsy was obtained using NGS panel testing [77]. Evaluation of MSI status by liquid biopsy has been reported [78]. Additionally, germline alterations, such as BRCA1/2 mutations in PDAC, were also detected by liquid biopsy. About 3% of germline-assumed alterations in 10,000 patients with PDAC were detected by analyzing ctDNA [79]. Compared with variant allele frequency of somatic mutations, that of germline mutations was ~50%; thus, it is relatively easy to distinguish them. From this viewpoint, ctDNA analysis by liquid biopsy may soon replace conventional CGP for histological analysis. However, there is insufficient evidence to initiate a prospective study on precision medicine based on the biomarkers of ctDNA alterations.

In Japan, the GOZILA trial, wherein ctDNA alterations of unresectable advanced gastroenterological cancers were screened using NGS, has been conducted. This trial is one of the largest screening projects for ctDNA assessment, the target of which was ~2000 patients. Furthermore, some physician-led clinical umbrella/basket trials, where drugs are chosen on the basis of ctDNA alterations, are being conducted to provide more evidence [80,81].

## 7. Conclusions

Precision medicine for advanced unresectable hepatobiliary and pancreatic cancers is reviewed focusing on clinical trial outcomes in this article. Although precision medicine methods have been improved to achieve better diagnostics of various tumor types, potentially effective drugs are prescribed to only ~10% of patients based on CGP tests in most types of cancers [82,83,84]. Not all actionable mutations concerned with oncogenesis are druggable, and not all druggable mutations can be treated with the drug because of the lack of adequate clinical trials. Many patients do not have adequate opportunities to be assessed using the precision medicine approach and treated with potentially effective drugs. From this viewpoint, it is crucially important to promote tumor-agnostic universal basket trials based on genomic information that leads to the development of new drugs to gain further advances in precision medicine.

Despite the recent progress of targeted inhibitors and ICIs, the overall outcomes of hepatobiliary and pancreatic cancers remain poor mainly due to the treatment resistance caused by genetic and non-genetic events. For example, a number of studies identified various genetic aberrations associated with resistance to each targeted therapy in various tumors, including FGFR2 fusion [85,86,87], HER2 amplification [88,89,90], BRAF V600E mutation [91,92,93], BRCA1/2 mutations [94,95,96], and NTRK fusions [97,98,99]. Our understanding of resistance mechanisms to targeted therapies would allow the development of rational approaches for hepatobiliary and pancreatic cancers in a patient-specific manner.

## Figures and Tables

**Table 1 cancers-14-03674-t001:** Therapy-targeted gene alterations and clinical trial treatment results for hepatobiliary and pancreatic cancers.

Cancer Type/Genes	Alterations	Prevalence	Drugs	Study	Efficacy
Biliary tract cancer (BTC)		[1]			
*IDH1*	Mutations	IHCCA 16%	Ivosidenib	Phase3 ClarIDHy [8,9]	ORR 2% (3/124), DCR 53% (66/124), ☆ PFS 2.7 m (HR0.37), MST 10.3 m (HR 0.49)
*FGFR2*	Fusions	IHCCA 11%	Pemigatinib Infigratinib	Phase2 FIGHT-202 [10] Phase2 [11]	☆ ORR 36% (38/107), DCR 82% (88/107) ☆ ORR 19% (9/48), DCR 83% (40/48)
*ERBB2*	Amplification	GBCA 16% EHCCA 11% IHCCA 3%	Trastuzumab + Pertuzumab T-DM1	Phase2a MyPathway [12] Phase2 NCI-MATCH [13]	☆ ORR 29% (2/7), DCR 86% (6/7) ☆ ORR 0% (0/3), DCR 100% (3/3)
*BRAF* ^V600E^	Mutations	5%	Dabrafenib + Trametinib	Phase2 ROAR [14]	☆ ORR 47% (20/43)
**Pancreatic cancer (PDAC)**					
*BRCA1/2*	Mutations	6% [15]	Olaparib	Phase3 POLO [16]	ORR 23% (18/78), ☆ PFS 7.4 m (HR0.53), MST 18.9 m (HR0.91)
*KRAS* ^G12C^	Mutations	1.7% [17]	Sotorasib	Phase1/2 CodeBreaK 100 [18]	ORR 9.1% (1/11), DCR 82% (9/11)
*NRG1*	Fusions	0.5% [19]	Zenocutuzumab	Phase1/2 eNRGy [20]	☆ ORR 42% (5/12), DCR 92% (11/12)
**Hepatocellular carcinoma (HCC)**					
*TERT*	Mutations	50% [4,5]	-	-	-
*TP53*	Mutations	40% [4,5]	-	-	-
Wint/βcatenin signaling pathway	Mutations	40–50% [21]	-	-	-

Abbreviations: BTC, biliary tract cancer; IHCCA, intrahepatic cholangiocarcinoma; GBCA, gallbladder cancer; EHCCA, extrahepatic cholangiocarcinoma; ORR, objective response rate; PFS, progression-free survival; MST, mean survival time; HR, hazard ratio; T-DM1, ado-trastuzumab emtancine; PDAC, pancreatic ductal adenocarcinoma; MSI-H, microsatellite instability-high; HCC, hepatocellular carcinoma; and M, months. ☆ Primary endpoint.

**Table 2 cancers-14-03674-t002:** Tumor-agnostic biomarkers and treatment results for hepatobiliary and pancreatic cancers.

	MSI-H	TMB-H	*NTRK*	
**Alterations**	Mutations of *MLH1*, *MSH2*, *MSH6*, *PMS2*	-	Fusions	
**Prevalence** Whole cancers Hepatobiliary pancreatic	[52] 5% Around 2%	[53] 13% 4%	[54] 0.26% (87/33997) BTC 0.34% (2/787), PDAC 0.25% (5/1492)	
**Drugs**	Pembrolizumab	Pembrolizumab	Larotrectinib	Entrectinib
**Study**	Phase 2 KEYNOTE-158	Phase 2 KEYNOTE-158	Integrated analysis of three phase 1/2	Integrated analysis of two phase 2 and one phase 1
**Efficacy** Whole cancers Hepatobiliary	[55] ☆ ORR 34.3% (80/233) BTC 41% (9/22), PDAC 18% (4/22)	[56] ☆ ORR 29.4% (30/102) no data	[57] ☆ ORR 79% (121/153) ORR of BTC 50% (1/2); PDAC 50% (1/2)	[58] ☆ ORR 57 % (31/54) ORR of BTC 100% (1/1); PDAC 67% (2/3)

Abbreviations: BTC, biliary tract cancer; ORR, objective response rate; PDAC, pancreatic ductal adenocarcinoma; and MSI-H, microsatellite instability-high. ☆ Primary endpoint.

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
