# Peer review of "Precision Medicine of Hepatobiliary and Pancreatic Cancers: Focusing on Clinical Trial Outcomes"

_cancers, 2022, doi:10.3390/cancers14153674_

Round 1
Reviewer 1 Report
Overall, this is a well-researched and composed review article addressing a timely and important topic. Minor comments that I think would strengthen this paper include:
1) Most of the works cited in this review are from Western studies. I think adding studies from other geographic areas (like the GOZILA study they cited), particularly East Asia (Chinese, Japanese, South Korean) could benefit readers as this is where the authors are from.
2) The authors should consider adding studies addressing resistance mechanisms to targeted therapies, including both observational/clinical and mechanistic/pre-clinical studies.
3) There is a newly published work from Japan (https://jamanetwork.com/journals/jamaoncology/fullarticle/2791277) extending germline BRCA-associated cancer types to gastric and biliary cancers. I think this is relevant and the authors may consider including it into their review.
4) There is a recent FDA approval for BRAF/MEK inhibitors for tissue agnostic indications (https://www.precisiononcologynews.com/cancer/novartis-nets-tumor-agnostic-approval-tafinlar-mekinist-combo#.YrYIaXbMLq4) This should be added to the tissue agnostic section.
5) The authors should consider the impact of CHIP (clonal hematopoiesis of indeterminate potential) mutations on the detection and interpretation of liquid biopsy findings (both germline and somatic)
Author Response
We thank the reviewer for his/her thoughtful comments, which have allowed us to improve the paper. We have addressed the reviewer’s comments through the following changes to the manuscript.
1) Most of the works cited in this review are from Western studies. I think adding studies from other geographic areas (like the GOZILA study they cited), particularly East Asia (Chinese, Japanese, South Korean) could benefit readers as this is where the authors are from.
Reply:
According to the reviewer’s suggestion, we cited additional papers from East Asia (references [70, 80, 83, 84]).
2) The authors should consider adding studies addressing resistance mechanisms to targeted therapies, including both observational/clinical and mechanistic/pre-clinical studies.
Reply:
According to the reviewer’s suggestion, we added studies (references [85-99]) and description (page9, lines 390-397) about resistance mechanisms to target therapies in the revised manuscript.
3) There is a newly published work from Japan (https://jamanetwork.com/journals/jamaoncology/fullarticle/2791277) extending germline BRCA-associated cancer types to gastric and biliary cancers. I think this is relevant and the authors may consider including it into their review.
Reply:
According to the reviewer’s suggestion, we modified the sentence (page5, lines 189) and added the paper in the revised manuscript (reference [38]).
4) There is a recent FDA approval for BRAF/MEK inhibitors for tissue agnostic indications (https://www.precisiononcologynews.com/cancer/novartis-nets-tumor-agnostic-approval-tafinlar-mekinist-combo#.YrYIaXbMLq4). This should be added to the tissue agnostic section.
Reply: According to the reviewer’s suggestion, we added the sentence in the revised manuscript (page7, lines 283-285).
5) The authors should consider the impact of CHIP (clonal hematopoiesis of indeterminate potential) mutations on the detection and interpretation of liquid biopsy findings (both germline and somatic).
Reply:
According to the reviewer’s suggestion, we added description about CHIP in the revised manuscript (page9, lines 355-360).
Reviewer 2 Report
This is a review of outcome of the clinical trials where targeted therapy has been applied to hepatobiliary and pancreatic cancer patients. Some are more effective than others and the response varies in patient population and so does the adverse outcome within one trial. In summary section, the authors needs to expand what is learnt from these trial outcomes beyond what they have currently mentioned.
Author Response
We thank the reviewer for his/her thoughtful comments, which have allowed us to improve the paper. We have addressed the reviewer’s comments through the following changes to the manuscript.
1) This is a review of outcome of the clinical trials where targeted therapy has been applied to hepatobiliary and pancreatic cancer patients. Some are more effective than others and the response varies in patient population and so does the adverse outcome within one trial. In summary section, the authors need to expand what is learnt from these trial outcomes beyond what they have currently mentioned.
Reply: According to the reviewer’s suggestion, we added description about the problem of targeted therapy needing to be explored in Summary section of the revised manuscript (page9, lines 390-397).